# The Effect of Managerial Myopia on the Adjustment Speed of the Company's Financial Leverage towards the Optimal Leverage

Vahab Rostami [1,*], Hamed Kargar [1] and Mahdis Samimifard [2]

[1] Department of Economics and Administrative Sciences, Payame Noor University, Tehran 19395-4697, Iran
[2] Department of Accounting, Zanjan University, Zanjan 45371-38791, Iran
[*] Correspondence: vahab.rostami@pnu.ac.ir

**Abstract:** The adjustment speed of financial leverage indicates the movement of companies towards the optimal capital structure, and clearly shows the financing policies of companies. The importance of optimal leverage is such that the growth and survival of companies depend on this factor. This study investigates the effect of managers' myopia on the adjustment speed of financial leverage toward optimal leverage. The current research is applied, and from the methodological point of view, the correlation is a causal type (retrospective). The statistical population of the research includes all the companies admitted to the Tehran Stock Exchange between 2011 and 2020, and using the systematic elimination sampling method, 124 companies were selected as the research sample. The research results showed that the myopia of managers has an opposite effect on the adjustment speed of financial leverage, so in companies with myopic managers, the adjustment speed of financial leverage decreases towards optimal leverage.

**Keywords:** adjustment speed of financial leverage; optimal leverage; managerial myopia

## 1. Introduction

In today's world, organizations and business enterprises operate in an environment of competition in the capital market, and the manager is considered to be the heart of every organization. Despite a company's facilities, technology, and optimal workforce, a poorly performing manager will result in the company's failure to achieve its goals (Salehi et al. 2022). Choosing efficient managers is one of the most critical concerns of companies nowadays. In this regard, it can be seen that the behavior of managers caused by their personality traits will affect the performance of companies (Daliri et al. 2020). Chief executive officers (CEOs) play a vital role in the company's strategic decision-making. Over the past decades, the responsibilities of CEOs have increased significantly. CEOs today must participate in the company's decision-making process, especially in financial matters. In certain companies the CEO is required to make financial decisions. At the same time, certain other financial decisions of companies result from the satisfaction of senior managers. Managers may not always choose debt options that maximize shareholder value, but prefer to limit the use of debt for their own interests (Zimon et al. 2022). This conflict creates additional agency costs and ultimately leads to the company's poor performance (Zahedi et al. 2022). CEOs often protect their interests when making financial decisions about companies (Naseem et al. 2019; Seifzadeh et al. 2022; Salehi and Zimon 2021). According to previous studies on people's personality traits, the results show that people's behavior largely depends on their personality traits, and one of these influencing factors is the phenomenon of myopia among managers (Almaleki et al. 2021). The capital market and the performance of companies in this market are influenced by a variety of factors (Kieschnick and Moussawi 2018). One of the critical influential factors is the decisions of company managers, who are the final decision-makers in executive matters. Managerial myopia is one such critical factor that can significantly impact the decisions of investors and managers,

and can deprive the market of the required efficiency. This factor results in decisions being made based on the market's short-term performance, with no plan considered for future performance. Myopic managers lose long-term benefits by increasing stock prices and short-term benefits (Chen et al. 2015; Salehi et al. 2021). Myopia, with a tendency towards short-term goals and moving towards meeting these goals by changing operating methods, will benefit the company's business process and economic profit in the short term. If investors are aware of profit inflation, they will probably consider the profit information worthless (Moradi and Bagheri 2014). The most crucial duty of managers in achieving companies' goals is to optimally combine financial resources in the capital structure, so that the company achieves its optimal leverage as quickly as possible (Oino and Ukaegbu 2015). The adjustment speed of financial leverage indicates the movement of companies towards the optimal capital structure, and clearly shows the financing policies of companies. The importance of optimal leverage is such that the growth and survival of companies depend on this factor, and affects the risk and expected returns of companies (Akbari et al. 2019). The speed of adjusting the leverage should be included in the company's strategic goals. Attention should be paid to the fact that the distance between the actual financial leverage and the target leverage will be as close as possible. From this point of view, myopic managers only focus on the profits and short-term performance of the company. According to behavioral finance theories, this may reduce the speed of adjusting the financial leverage towards the target leverage, so it is necessary to answer the question of whether managerial myopia affects the adjustment speed of the leverage. Considering the research gap created, the lack of definitive findings, the vital role of managers in companies, the increase of joint-stock companies, and the optimal financial leverage that guarantees the survival of companies, the necessity of the current research becomes more apparent. In this paper, firstly the development of the theoretical foundations, hypotheses, and experimental foundations are presented, followed by the research methodology and operational definitions variables, and finally, the findings and conclusions of the research are presented.

## 2. Theoretical Principles of the Study

In today's competitive world, companies that establish plans with a long-term vision remain stable. Many of the policies adopted by managers appear to be promising because they show good effects in the short term. It is certain that in such a highly competitive market, companies that establish plans with a long-term perspective remain successful (Naseem et al. 2019; Salehi and Ghasempour 2021). Managers in organizations and institutions are the central pillar of decision-making, especially senior managers who will determine the fate and future of the company with the final decision. Therefore, various factors can influence the decisions made by managers. In the field of behavioral finance studies we can refer to myopia, overconfidence, and the ability of managers as influencing factors. Each of these personality traits can affect the continuation of the company's activities. Managerial myopia is one such feature that can limit the company's future, its long-term activities and future investments (Arianpoor and Mehrfard 2022; Mohammadi et al. 2018).

Myopia means overestimating short-term earnings and underestimating long-term earnings by investors who are active in the market. In markets without efficiency characteristics, investors and decision-makers determine the value of the company only based on what happened in the past or in the short-term future, and they attach less importance to the potential capabilities of the company in the distant future (Abarbanell and Bernard 2000; Faysal et al. 2020). Myopia is a form of permanent bias or prejudice that has been proven in how people and organizations view the world. Another perspective is on managerial myopia as a form of bias that affects investment decisions. One of the manager's most essential duties in achieving the company's goals is to combine financial resources in the capital structure optimally. In their financing methods, according to risk and return, they choose an optimal combination of debt and equity to minimize financing costs, thereby maximizing shareholders' interests (Oino and Ukaegbu 2015). One of the ways to achieve this goal is to reduce the risks faced by the company and consequently increase the com-

pany's efficiency. Therefore, it is necessary to discuss the speed of adjusting the financial leverage towards the optimal leverage, because one of the main reasons for the failure of companies is insufficient investment and inappropriate financing. The speed of adjustment of financial leverage is when the company adjusts its capital structure and moves toward the optimal leverage that it has already targeted and achieved (Moradi and Abad 2021; Seifzadeh et al. 2021). Company managers play an essentially supervisory role in this field; risk-averse managers prefer lower levels of debt to avoid the risk of bankruptcy (Rostami et al. 2022). The optimal financial leverage is said to be a combination affected by the correct and targeted use of financial resources, and obtaining a reasonable and suitable return with the risk of these resources. The speed of moving towards the optimal ratio depends on various factors, of which the cost of adjusting the financial leverage is one of the most critical factors. The most crucial issue in this context is the speed of their movement toward the target lever. They often take measures to adjust the capital structure, and say that the benefits of capital structure adjustment will outweigh its costs (An et al. 2021). Optimal financial leverage is the ratio of debt to assets in which the company's value is at a maximum, and the total cost of capital is at a minimum. Managers always seek to choose the optimal capital structure due to the company's status and performance to the stakeholders (Chang et al. 2014). Therefore, the optimal capital structure is determined, as the target capital structure companies are trying to put their actual capital structure within its limits. Based on theoretical concepts, managers should plan the company's optimal capital structure (Nazemi Ardakani and Zare 2015). CEOs today must participate in the company's decision-making process, especially in financial matters. In certain companies, the CEO makes financial decisions; while in others, the financial decisions of companies are the result of the satisfaction of senior managers. Managers may not always choose debt options that maximize shareholder value, but prefer to limit the use of debt to their own interests. This conflict creates agency costs and ultimately leads to poor company performance (Naseem et al. 2019). CEOs often protect their own interests when making financial decisions about companies. This practice is very common in developing countries when a company grows rapidly; there is a tendency for high debt-to-equity ratios due to the CEO's reluctance to issue shares. A serious selection issue regarding the source of financing among retained earnings is reported as debt. Some theoretical studies have shown that CEO characteristics may affect firm performance. For example, female CEOs may cause less leverage, which can lead to bias in the capital allocation process, as female CEOs are more risk-averse than their male counterparts (Faccio et al. 2016).

Meanwhile, myopic managers tend to choose between behaviors that cost the loss of long-term investment and increase short-term benefits. These behaviors aim to increase the company's stock price in the short term. Managerial myopia challenges the ability of financial markets as well as the value of companies. If markets are efficient, managers will refrain from myopic behavior because investors perceive that short-sighted behavior reduces the ability of companies to generate long-term profits (Moradi et al. 2016). This may affect the speed of adjustment of financial leverage towards optimal leverage according to the stated content, and also the fact that myopic behavioral theory focuses on the short-term performance of the company and the speed of adjustment of leverage towards optimal leverage. It should be in the strategic plans and should be one of the general and permanent goals of the company, so that the managers' myopia can reduce the speed of lever adjustment. Therefore, according to the stated contents, the research hypothesis is presented as follows:

**Research Hypothesis.** *Managers' myopia reduces the speed of adjustment of financial leverage towards optimal leverage.*

*The Empirical Background of the Research*

Within a study entitled *Product market threats and leverage adjustment*, Do et al. (2022) stated that the effect of product market threats on leverage adjustment is more evident for

companies that have poor governance quality and are exposed to product market threats, to the extent that achieving the target capital structure finally increases the value of the company. Morais et al. (2022) showed that companies with zero leverage actively adjust to the target debt ratio. Only when the analysis is restricted to financially constrained firms is there a significant difference between the two groups; otherwise, there is no difference. During the 2008 financial crisis, zero leveraged firms adjusted their capital structure significantly faster (46.8%) than leveraged firms (25.6%) and compared to non-crisis years (21.6%).

In a research study titled *Capital structure adjustment in emerging markets*, Arikawa and Hoang (2022) stated that the speed of capital structure adjustment was investigated using a partial adjustment model. The results showed that the speed of adjustment in emerging markets is very slow, and Vietnamese companies do not adjust their capital structure with high flexibility toward optimal value. In addition, Vietnamese companies mainly use debt as external financing. Rostami et al. (2022) stated that the speed of adjustment of financial leverage indicates the movement of companies towards an optimal capital structure. The importance of optimal financial leverage is such that the growth and survival of companies depend on this factor, and affects the risk and expected returns of companies. The results of the regression test showed that risk management has a direct effect on the speed of adjustment of financial leverage. In addition, risk management in a company's growth period has a direct effect on the speed of adjustment of financial leverage with an increasing factor. Still, in the maturity period, risk management does not affect the speed of adjustment of financial leverage. Additionally, risk management in a company's period of decline with a decreasing and negative coefficient, has an inverse effect on the speed of adjustment of financial leverage. Therefore, according to the obtained results, by managing the risks facing the company, companies can move towards the optimal financial leverage faster. This effect is reduced in the company's life cycle transition stages. Vo et al. (2021) stated that, on average, companies tended to adjust their capital structure more quickly after the COVID-19 outbreak. In addition, in countries where COVID-19 is causing more severe damage, companies adjust their target leverage faster than those in less affected countries. In a research paper entitled *Foreign ownership and the speed of adjustment of financial leverage*, An et al. (2021) stated that there is a positive relationship between foreign institutional ownership and the speed of adjustment of companies' leverage. Foreign institutional investors have an essential regulatory role in reducing agency conflicts between shareholders and managers. Assadi et al. (2021) showed that the debt ratio is higher than the target debt ratio, and companies with financial surplus tend to reduce the debt ratio. In addition, companies under concentrated leverage (less competitive) will have less desire to adjust their debt ratio and increase debt. In companies with a high dynamic leverage, the drive to reduce the debt ratio is stronger, and the movement toward the target leverage occurs faster. Daliri et al. (2020) stated that the relationship between managerial myopia and sustainability of the competition and reporting is inverse (negative) and significant. Additionally, the relationship between managerial myopia and social sustainability, social impact, and commitment is negative and significant. Still, it has a positive and meaningless relationship with social interests. Finally, the lack of myopia in managers and sustainable performance, a combination of two social and economic components, is negative and significant. Fitzgerald and Ryan (2019) stated that small companies with high growth and low dividends paid leverage faster than large companies with low growth and high dividends. Tehrani and Delshad (2018) expressed that managerial myopia positively and significantly affects future financial performance. Chintrakarn et al. (2016) stated that companies with weaker corporate governance have more myopic management. In addition, strong corporate governance increases the motivation of executive managers to implement research and development projects and long-term investments, thus reducing managerial myopia. In a study, Moradi et al. (2016) concluded that short-termism or managerial myopia has caused the fundamental factors and long-term horizon of investing in the stock market to be forgotten, which has replaced the acquisition of daily returns. The statistical



results confirm the existence of the phenomenon of myopia in the Iranian capital market and show its inefficiency. Chen et al. (2015) measured the influence of institutional investors on the phenomenon of managerial myopia in emerging markets (Taiwan), and expressed that managers should reduce research and development costs in emerging markets in order to achieve short-term revenue goals. While the institutional investors inside an organization intensify managerial myopia, in contrast, external institutional investors act as a shield, preventing managerial myopia of research and development costs.

## 3. Research Methodology

This paper is an applied and, from a methodological point of view, correlation-causal type (retrospective). The statistical population studied in this research includes all listed companies on the Tehran Stock Exchange from 2011 to 2020. With the following conditions, the companies that were admitted to the Tehran Stock Exchange have been selected as the study sample to make the information comparable. The end of the financial year of the companies should be the end of March. Companies should not have changed the financial period under review during the (10-year) period. Information about the selected variables in this research should be available. Companies should not be affiliated with banks, insurance companies and investment companies. Finally, 124 companies have been selected as the final sample of the research. Data analysis has been undertaken using the combined data method and panel data approach, Eviews 12 software, and the powerful standard tool to test the hypotheses.

### 3.1. Regression Model of Research

$SL_{i,t} = \beta_0 + \beta_1 MYO_{i,t} + \beta_2 STATE_{i,t} + \beta_3 Size_{i,t} + \beta_4 ROA_{i,t} + \beta_5 PMC_{i,t} + \beta_6 INST_{i,t} + \beta_7 Growth_{i,t} + \beta_8 CASH_{i,t} + \beta_9 BTM\ ratio_{i,t} + \beta_{10} Board\ ind_{i,t} + \varepsilon_{i,t}$

### 3.2. Operational Definitions of Research Variables

The independent variable of the research is managerial myopia (MYO).

According to the research of Anderson and Hsiao (1982) and Moradi and Bagheri (2014), managers with a short-term horizon were calculated in the following way. It is a two-valued variable (0 and 1), which is equal to the number (1) if the managers are myopic; otherwise, it will be (0). The method of calculating the variable of managers with a short-term horizon is as follows:

It is expected that when companies achieve significant financial success, they will have the opportunity and resources to invest in future long-term assets. Therefore, companies that simultaneously report higher than expected returns and less than normal marketing and research and development expenses will be subject to the characteristic of myopic management. To identify and determine myopic companies, it is first necessary to estimate the expected level of return on assets, marketing cost and research and development cost for each company in each period. In this regard, following Anderson and Hsiao (1982), the following equations have been used:

$$ROA_{i,t} = \beta_0 + \beta_1 ROA_{i,t-1} + \varepsilon_{i,t}$$

$$MKTG_{i,t} = \beta_0 + \beta_1 MLTG_{i,t-1} + \varepsilon_{i,t}$$

$$R\&D_{i,t} = \beta_0 + \beta_1 R\&D_{i,t-1} + \varepsilon_{i,t}$$

In these equations, ROA, MKTG, and R&D returns on assets (net profit divided by total assets) marketing expense, and R&D expense for the company i in period t. It is recalled that marketing expenses and R&D expenses are extracted from the companies' disclosed explanatory notes. After calculating the estimated values of return on assets, marketing cost and research and development cost using the above models, the predicted values obtained from the model are compared with the actual values. According to the prediction error of these three models, the companies are divided into four main groups, as described in Table 1.

**Table 1.** The Statistical sample grouping.

| Group 1 | Group 2 | Group 3 | Group 4 |
|---|---|---|---|
| The difference in return on assets | The difference in return on assets | The difference in return on assets | The difference in return on assets |
| Predicted with positive real | Predicted with positive real | Predicted with positive real returns | Predicted with negative real |
| The difference in marketing and research and development costs | Only the difference of one between negative marketing or research and development costs | The difference in marketing and research and development costs | - |

Among these groups, group 1 is represents companies with myopic management. Despite the company's positive performance and asset yield increase, marketing, research, and development costs have decreased in this group. Finally, a dummy variable is used to show myopic management, and 1 is given for the companies placed in group 1; otherwise, the number is zero (Moradi and Bagheri 2014).

*3.3. The Dependent Variable of the Research: The Adjustment Speed of Financial Leverage (SL)*

In many capital structure studies, the partial adjustment model is used to measure the speed of adjustment (Flannery and Rangan 2006; Öztekin 2015). In the partial adjustment model, both actual and optimal leverage should be measured in the first step. Still, since optimal leverage cannot be measured directly, its value must be obtained by replacing other variables. In this research, those obvious characteristics of the company that influence financing decisions are considered; other characteristics, such as the economic situation and unobservable (uncontrollable) effects that affect financing decisions and are not easily measured, are considered as the error of the estimator. The optimal leverage is estimated using the following model:

$$L^*_{it} = \beta' x_{it} + u_{it} \tag{1}$$

where

$L^*_{it}$ optimal leverage; $x_{it}$ is a vector of the characteristics of the i-th company at time t, which is related to the benefits and costs of activity under different leverage ratios, $\beta'$ is the estimated coefficient of this vector and $u_{it}$ is the error component of the model (Dang et al. 2014; Rostami et al. 2022).

To select the characteristics of the company, the most used variables are used in the research on the company's capital structure (Rostami et al. 2022).

1. Financial deficit: dividends paid plus net cash from investing activities plus changes in working capital minus cash from operating activities, divided by the company's market value.

2. Growth opportunities: dividing the market value of the equity by the book value of the company's total assets.

3. Income fluctuations: the absolute value of the difference in each period's income from the company's 5-year average income divided by the 5-year average income.

4. Profitability: the ratio of annual profit before interest and taxes to its total assets at the end of the year.

5. Tangible fixed assets: dividing fixed assets by total assets.

6. Company size: natural logarithm of assets.

7. firm age: the natural logarithm of the company's year of establishment to the year of the research time horizon.

By replacing the characteristics of the company, in model 1, the optimal leverage will be obtained by the following model:

$$L^*_{it} = \beta_1 SIZE_{it} + \beta_2 EBIT_{it} + \beta_3 GROW_{it} + \beta_4 EV_{it} + \beta_5 AGE_{it} + \beta_6 FA_{it} + \beta_7 FIMB_{it} + u_{it} \tag{2}$$

where

L*$_{it}$ is optimal leverage; SIZE, company size; EBIT, profitability; GROW, growth opportunities; EV, income volatility; AGE, company age; FA, tangible fixed assets; FIMB, fiscal deficit and u$_{it}$ is the error component.

As mentioned, the partial adjustment model is used to obtain the optimal financial leverage adjustment speed. In this research, the partial adjustment model of Fama and French (2002) is used as follows:

$$\Delta L_{it} = \lambda(L^*_{it} - L_{it-1}) + v_{it} \tag{3}$$

where

$\Delta L_{it}$ is the difference between the actual leverage of the current year and the actual leverage of the previous year; L*$_{it}$, is optimal leverage; L$_{it-1}$ is the real leverage of the previous year; $\lambda$ is the accrual rate, and v$_{it}$ is the one-sided error component that includes the firm unique fixed effects u$_{it}$ model (2) and is the error component (e$_{it}$).

This model allows the company to reduce the gap between its actual leverage and its target leverage by 1 every year. The range of coefficient 1 is between zero and one, and a value close to one indicates a higher adjustment speed and vice versa. For the final calculation of the adjustment speed of the above two patterns, the following pattern is obtained by merging:

$$L_{it} = \varnothing_1 SIZE_{it} + \varnothing_2 EBIT_{it} + \varnothing_3 GROW_{it} + \varnothing_4 EV_{it} + \varnothing_5 AGE_{it} + \beta\varnothing_6 FA_{it} + \varnothing_7 FIMB_{it} + (1-\lambda)L_{it-1} + v_{it} \tag{4}$$

where

$\varnothing\_1$ to $\varnothing\_7$ is equal to $\lambda\beta'$; SIZE, company size; EBIT profitability; GRO, growth opportunities; EV, earnings volatility; AGE, company age; FA, tangible fixed assets; FIMB, fiscal deficit; $\lambda$, the rate of adjustment and L$_{it-1}$, the real leverage of the previous year.

The above model states that managers usually adopt strategies that reduce the gap between their current and desired capital structure positions. In addition, this equation assumes that all companies adjust their capital structure at the same rate (Fama and French 2002); Therefore, by subtracting the estimated coefficient for L$_{it-1}$ from the number one, the speed of lever adjustment will be obtained (the estimated coefficient was obtained using rolling regression facilities).

*3.4. The Research Control Variables*

Return on assets (ROA): To calculate this variable, dividing net profit by total assets is used.

Company size (SIZE): The natural logarithm of the sum of assets was used to calculate this variable.

Market competition (PMC): This variable is calculated by the Herfindahl-Hirschman Index (the square root of the company's sales revenue divided by the total sales revenue of the companies in the company's industry).

Company growth (BTMratio): To calculate this variable, dividing the market value of capital by the book value of capital at the end of the financial year is used.

Growth: Sales of the period minus sales of the previous period divided by sales of the previous period.

Independence of the board of directors (Boardind): the ratio of non-executive members to the total number of members.

Cash: the ratio of operating cash to total assets.

Political relations (SATE): If the largest shareholder is a government or government-affiliated company, it will be (1); otherwise, (0).

Institutional investors (Inst): Investors such as banks, insurance and investment companies, and individuals and companies who own more than 5% of the company's shares are called institutional shareholders, and institutional investors have used the percentage of shares held by this group.

## 4. Research Findings

### 4.1. Descriptive Statistics of Research Variables

The main centrality index is the mean according to Table 2, which indicates the distribution's balance point and center of gravity and is a good indicator to show the centrality of the data. For example, the average value for the financial leverage adjustment speed variable is equal to (0.53), which shows that most data are concentrated around this point. In general, dispersion parameters are a measure to determine the degree of dispersion from each other, or their degree of dispersion compared to the average. One of the most important dispersion parameters is the standard deviation. The value of this parameter for institutional ownership is equal to 30.84%, and the liquidity ratio is 0.13%, which shows that these two variables have the highest and lowest standard deviation, respectively. Minimum and maximum also indicate the minimum and maximum in each variable. For example, the largest company size value is 19.77%.

**Table 2.** The Descriptive statistics of quantitative research variables.

| Variable | Sign | Mean | Maximum | Minimum | Standard Deviation | Kurtosis | Skewness |
|---|---|---|---|---|---|---|---|
| Leverage adjustment speed | SL | 0.530 | 0.980 | 0.004 | 0.260 | −0.260 | 2.110 |
| Liquidity ratio | Cash | 0.120 | 0.510 | −0.120 | 0.130 | 0.750 | 3.580 |
| Board independence | Board ind | 0.650 | 1.000 | 0.000 | 0.170 | −0.210 | 3.090 |
| Firm growth | BTM ratio | 5.650 | 15.870 | 1.000 | 4.790 | 1.080 | 2.870 |
| Sales growth | Growth | 0.290 | 1.320 | −0.390 | 0.380 | 0.740 | 3.430 |
| Institutional ownership | INST | 60.280 | 98.110 | 0.000 | 30.840 | −0.850 | 2.400 |
| Product market competition | PMC | 0.0750 | 0.980 | 0.000 | 0.210 | 3.510 | 14.280 |
| Return on assets | ROA | 0.130 | 0.590 | −0.170 | 0.140 | 0.760 | 3.620 |
| Firm size | SIZE | 14.480 | 19.770 | 11.030 | 1.490 | 0.870 | 4.200 |

### 4.2. Descriptive Statistics of Qualitative Research Variables

The myopia variables of the managers to the political communication of the company are qualitative variables, whose frequency distribution tables are presented as follows.

As seen in Table 3, the total number of companies under investigation is equal to 1240 cases, of which 441 companies have myopic managers, and 799 companies have 64.44 percent of the year. In addition, 503 company years, equivalent to 40.56% of the companies, have political relations, and 737 company years, equivalent to 59.44%, have no political relations.

**Table 3.** The frequency distribution of myopia variable of managers and political communication.

| Variable | Sign | Description | Frequency | Frequency Percentage |
|---|---|---|---|---|
| Myopic managers | MYO | 1 | 441 | 35.560 |
| Non-myopic managers | MYO | 0 | 799 | 64.440 |
| Companies with political connections | STATE | 1 | 503 | 40.560 |
| Companies without political connections | STATE | 0 | 737 | 59.440 |
| Total | - | - | 1240 | 100 |

### 4.3. Unit Root Test (Stability) of Variables

According to the results obtained in Table 4, it can be seen that the significance level of the variables in the significance test is less than 5%, and it indicates the significance of the variables.

**Table 4.** Levin, Lin and Chu test-quantitative research variables.

| Variable | Sign | Mean | Maximum | Results |
|---|---|---|---|---|
| Leverage adjustment speed | SL | −13.906 | 0.000 | Stationary |
| Liquidity ratio | Cash | −17.102 | 0.000 | Stationary |
| Board Independence | Board ind | −10.083 | 0.000 | Stationary |
| Firm growth | BTM ratio | −13.837 | 0.000 | Stationary |
| Sales growth | Growth | −14.563 | 0.000 | Stationary |
| Institutional ownership | INST | −13.458 | 0.000 | Stationary |
| Product market competition | PMC | −75.024 | 0.000 | Stationary |
| Return on assets | ROA | −12.980 | 0.000 | Stationary |
| Firm size | SIZE | −14.633 | 0.000 | Stationary |

*4.4. Tests Related to the Classical Hypothesis of Regression*

According to the results obtained in Table 5, it can be seen that the significance level of the F-Limer test is less than 5%, and the panel data model is accepted. Moreover, the Hausman test with a significance level of less than 5% confirmed the intercept's fixed effects. In addition, the tests of White and Breusch-Godfrey with a significance level below 5% indicate that there is serial correlation and heterogeneity of variance in the research model. This was finally resolved in the hypothesis test of the research using the standard powerful tool and command. The normalization of the error of the main hypothesis test model, with a significance level below 5%, shows that it does not follow the normal distribution level. This is not an important issue considering the application of the regression preconditions and the high level of data in the combined data.

**Table 5.** The results of the classic regression hypothesis test.

| Test | Test Statistic | Significance Level | Test Result |
|---|---|---|---|
| F. Limer | 1.840 | 0.000 | Admitting panel data pattern |
| Hausman | 27.763 | 0.002 | Fixed effects of intercept |
| White's test | 159.710 | 0.000 | Existence of heterogeneity of variance |
| Breusch-Godfrey test | 22.169 | 0.000 | The existence of serial autocorrelation |
| Normalization of model residuals | 35.788 | 0.000 | no normal distribution |

*4.5. The Result of the Research Hypothesis Test*

Research hypothesis: managerial myopia reduces the speed of adjustment of financial leverage towards optimal leverage.

The results of Table 6 show that the myopia variable of managers with a negative coefficient (−0.030) and a significance level below 5% (0.002) has an inverse relationship with the speed of financial leverage adjustment. Therefore, with the increase of managerial myopia, the speed of adjusting the leverage towards the optimal leverage decreases. Therefore, the research hypothesis is accepted at the error level of 5%. The control variables of return on assets, competition in the product market, institutional ownership, sales growth and company growth with a significance level below 5% have a significant relationship with the dependent variable of the research. The coefficient of determination is equal to 30%, which shows that the independent and control variables in the model have been able to explain 30% of the changes in the dependent variable. In addition, the Durbin-Watson value is equal to 2.28, which shows that there is no serial autocorrelation between the disruptive terms of the model. The test statistic with a significance level below 5% indicates that the research model fits well.

**Table 6.** The result of the research hypothesis test.

| $SL_{i,t} = \beta_0 + \beta_1\,MYO_{i,t} + \beta_2\,STATE_{i,t} + \beta_3\,Size_{i,t} + \beta_4\,ROA_{i,t} + \beta_5\,PMC_{i,t} + \beta_6\,INST_{i,t} + \beta_7\,Growth_{i,t} + \beta_8\,CASH_{i,t} + \beta_9\,BTM\ ratio_{i,t} + \beta_{10}\,Board\ ind_{i,t} + \varepsilon_{i,t}$ | | | | | | |
|---|---|---|---|---|---|---|
| **Dependent Variable: Adjustment Speed of Financial Leverage** | | | | | | |
| **Variables** | **Sing** | **Coefficient** | **Standard Error** | **t Statistic** | **Significance** | **VIF** |
| Managerial myopia | MYO | −0.030 | 0.010 | −3.060 | 0.002 | 1.150 |
| Political relation | STATE | 0.009 | 0.009 | 0.980 | 0.320 | 1.160 |
| Firm | SIZE | −0.002 | 0.003 | −0.780 | 0.430 | 1.200 |
| Return on assets | ROA | 0.410 | 0.040 | 10.040 | 0.000 | 1.630 |
| Product market competition | PMC | 0.067 | 0.019 | 3.430 | 0.000 | 1.020 |
| Institutional ownership | INST | −0.000 | 0.000 | −5.780 | 0.000 | 1.220 |
| Sales growth | Growth | −0.032 | 0.015 | −2.090 | 0.036 | 1.360 |
| Liquidity ratio | Cash | −0.020 | 0.035 | −0.570 | 0.560 | 1.380 |
| Firm growth | BTM ratio | −0.005 | 0.001 | −3.600 | 0.000 | 1.100 |
| Board independence | Board ind | 0.019 | 0.023 | 0.820 | 0.410 | 1.040 |
| Intercept | | 0.600 | 0.054 | 11.130 | 0.000 | - |
| The adjusted coefficient of determination | | | | 0.300 | | |
| Durbin-Watson | | | | 2.280 | | |
| F statistic | | | | 3.615 | | |
| Significance level | | | | 0.000 | | |

## 5. Conclusions

The final decision regarding the combination of financial leverage is one of the primary duties of company managers. The optimality of the financial leverage is of great importance because the optimal leverage can bring maximum value to shareholders. For this reason, companies should approach the level of the company's leverage towards optimal leverage as quickly as possible. Therefore, companies will consider the costs of adjusting the financial leverage and whether it is aligned with the company's interests; sometimes this is omitted due to the high costs. According to the balance theory, optimal leverage can create a kind of balance in between, the achievement of which will be the maximum value for the company. Meanwhile, the decisions of company managers, who are the final decision-makers in managing the company's affairs, can be influenced by their personality traits and thoughts. These thoughts can lead to managerial myopia. It is believed that such managers are focused on the company's short-term goals and earnings and do not pay attention to long-term planning, and such cases are not in line with their thoughts. According to the results, it was observed that the myopia of managers has an inverse effect on the speed of adjusting the financial leverage towards the optimal lever. In another way, it can be said that with the increase of managers' myopia, the speed of adjusting the lever towards the optimal lever will decrease. The time interval to reach optimal leverage will increase. Indeed, in such a situation, shareholders will not achieve their maximum benefits. Managers will seek to achieve specified short-term goals and stabilize their position in the company, unaware that they do not have a general long-term plan and will face problems in regards to the company's future interests. Usually, myopic managers are risk-averse and have no desire to use high debt. They try to increase stock returns in the short term, which can reduce the speed of lever adjustment towards optimal leverage. Identifying the optimal lever and moving towards it is one of the companies' strategic plans, and is far from the plans of myopic managers.

### 5.1. Discussion and Suggestions

The research results are somehow complementary to the research conducted on the influence of managers and company board of directors on the capital structure of companies. It is suggested that company managers consider short-term and long-term goals to balance advancing goals and maximize shareholder value. The company board of directors can successfully move the financial leverage towards optimal leverage by sufficiently monitoring

the decisions of myopic managers. It is suggested that future researchers investigate the impact of companies' risk management on the speed of adjustment of financial leverage.

*5.2. Research Limitations*

In the current research, the researcher did not deal with any specific and significant limitations. Only the discussion of sanctions and inflation, and their effects on the exchange rate and real earnings calculations of some companies, can be considered limitations of the research.

**Author Contributions:** Conceptualization, V.R.; methodology, H.K.; software, V.R.; validation, H.K.; formal analysis, M.S.; investigation, M.S.; resources, V.R.; data curation, V.R.; writing—original draft preparation, V.R.; writing—review and editing, V.R.; visualization, V.R.; supervision, V.R.; project administration, V.R.; funding acquisition, V.R. All authors have read and agreed to the published version of the manuscript.

**Funding:** The paper received no funding from any organizations.

**Data Availability Statement:** The data will be available at request.

**Acknowledgments:** Not applicable.

**Conflicts of Interest:** The authors declare no conflict of interest.

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
