# Peer review of "The Effect of Managerial Myopia on the Adjustment Speed of the Company’s Financial Leverage towards the Optimal Leverage"

_jrfm, doi:10.3390/jrfm15120581_

Round 1

Reviewer 1 Report

This paper is well written and comprehensive especially in the analysis of data and information presented.  I do wonder at time if highly statistical papers at times miss important aspects.  For example, in management decision making who are the stakeholders that need to be addressed - this paper seems to focus on the shareholder.  Clearly most important but this doesn't reflect other pressures on a company which may impact on it's ability to reach an optimal position.  It is equally fine to suggest this is a preferred position but exactly how is it identified at a point in time, and to what degree is it static in the, say, current environment we currently face?  The question of shortsightedness and managers, of we take a positive stance, might also relate to decisions  surrounding short term profitability particularly if linked to salaries/bonus payments.  Is speed of adjustment and 'optimum' tied to rewards?  What is the impact of current pressure in regard to sustainability (for example, climate change) - how might this impact on the optimum  Increasing levels of inflation?  Ease of manipulating financial leverage?  Competing pressures?  What is the nature of 'heavy competition'? Often managers are not in the role for the long term but looking toward the next better rewarded role - how might this impact?  What are the supervisory skills of board members?  There are many questions that need  to be asked - you have good information in your paper but also need to offer recognition to the fact there are limits to the insights offered.  Best of luck.

Author Response

Dear Reviewer,

Thank you very much for your good comments on the paper, further, the whole paper revised and retranslate by a native English speaker

Reviewer 2 Report

The paper studies the negative consequences of the managers' short-sightedness on the speed of adjustment of financial leverage toward optimal leverage.

The authors should eliminate citations from the abstract.

The paper contains adequate literature review on the investigated topic.

In Introduction, the authors should present the importance of the study, the relevance to the readers and the novelty of the paper. The short description of the paper sections presented at the end of the Introduction should be revised.

The methodology is appropriate considering the paper aim.

The results are clearly presented for the aim established. The Discussions section should be separated by the final conclusions.

In the Results and Discussions subsections, the authors should create a link between results achieved and results presented in the literature review at the beginning of the paper.

In Conclusions should be presented the limitations of the study and policy implications.

Author Response

Dear Reviewer,

Thank you very much for your good comment on the paper, the following issues has addressed in the current version:

The authors should eliminate citations from the abstract.

Response: Thank you very much the corrections has made in this part

The paper contains adequate literature review on the investigated topic.

Response: Thank you even in the current draft the most recent studies are included

In Introduction, the authors should present the importance of the study, the relevance to the readers and the novelty of the paper. The short description of the paper sections presented at the end of the Introduction should be revised.

Response: Thank you it is revised

The methodology is appropriate considering the paper aim.

Response: Thank you

The results are clearly presented for the aim established. The Discussions section should be separated by the final conclusions.

Response: Thank you it is revised

In the Results and Discussions subsections, the authors should create a link between results achieved and results presented in the literature review at the beginning of the paper.

Response: Revised

In Conclusions should be presented the limitations of the study and policy implications.

Response: The corrections has made on this part

Reviewer 3 Report

The reviewed study documents a negative consequence of the managers' short-sightedness on the speed of adjustment of financial leverage toward optimal leverage. While the idea is not entirely new, it is interesting to see it empirically examined in the Iranian setting. However, I still have a few concerns over the current study. I list them below.

1.     Writing

While I typically won’t list writing as my major concern on a study, the current manuscript is full of grammatic and other errors which hampers even the delivery of key messages in the paper. I urge the authors to polish the wiring with the help from professional editorial service.  

2.     The measurement

The authors’ key measure of adjustment speed of capital structure is based on Dong et al. (2014). But this paper is not cited. I have whether and how appropriate this measure is in the Iranian setting.

The other key measure, the short-sightedness, is based on Anderson and Hsiao (1982). But I don’t find discussion on short-sightedness in Anderson and Hsiao (1982).

With these, at this stage I cannot judge whether the empirical results are valid, let alone robust. I therefore urge the authors to elaborate more on these measures. 

Author Response

Dear Referee,

Thank you very much for your comment on the paper; your concerns have been incorporated into the current draft as follows:

  1. Writing

While I typically won’t list writing as my major concern on a study, the current manuscript is full of grammatic and other errors which hampers even the delivery of key messages in the paper. I urge the authors to polish the wiring with the help from professional editorial service.  

Response: Thank you very much for the whole paper was again translated by a native English speaker.

  1. The measurement

The authors’ key measure of adjustment speed of capital structure is based on Dong et al. (2014). But this paper is not cited. I have whether and how appropriate this measure is in the Iranian setting.

Response: Thank you, it is done.

The other key measure, the short-sightedness, is based on Anderson and Hsiao (1982). But I don’t find discussion on short-sightedness in Anderson and Hsiao (1982).

 Response: Thank you, it is done.

With these, at this stage I cannot judge whether the empirical results are valid, let alone robust. I therefore urge the authors to elaborate more on these measures. 

Response: Thank you; it is done.

Round 2

Reviewer 3 Report

I am satisfied with the revision.